# OpenReview forum: "An Analysis of Social Biases Present in BERT Variants Across Multiple Languages"
_NeurIPS.cc/2022/Workshop/TSRML — TSRML2022_

### Official Review · Reviewer_CkXQ · 2022-10-10
**Interesting mix of languages and bias queries; straightforward technical approach**

**Overall Rating:** 7

**Summary:**

The authors study sociolinguistic bias outside of or as a complement to the standard gender bias studies. The languages studied are English, Greek, and Persian. This is important as studies often focus on English or languages in Europe with strong cultural similarities to English-speaking countries. For their experimental results, a pseudo-likelihood is computed as a proxy for the probability of generating a sentence using BERT. This value is then used in a modified categorical bias score and studied in various prompts to naturally draw out bias in the models.


**Strengths:**

+ Very well-written
+ It was interesting to see some counter-intuitive results, such as the inversion in probabilities for “[MASK] are so hysterical” between the English and Persian language models.
+ The choice of languages reflects cultural differences and further cements the point that language models have to be studied in the context of culture, history, etc.
+ Code has been made available



**Weaknesses:**

- In a paper discussing bias, I think it is important for the authors to present their findings as objectively as possible, without bringing their own bias into the assumptions of why the results are what they are. This comment is in reference to Section 5.
- It is mentioned that “A perfectly unbiased model would produce equal probabilities regardless of the class mentioned.” This makes sense in the setting of this paper, but what about settings where bias would be important? For example, given the Russia-Ukraine conflict, it is reasonable for a language model to say that “people from Ukraine need your food donations” or something to that effect as opposed to having some uniform distribution over which countries need aid. This is not really a weakness of the paper as they have made their setting clear, but I was wondering if the authors have any comments on this front.
- Minor grammatical typo: “In our investigation, We” -> "In our investigation, we"
- The metrics used for computing bias are fairly straightforward adaptations of existing work





**Overall Recommendation:**

I recommend this paper for acceptance, though I will not champion it. It is very well-written and studies a good mix of languages, though the technical approach is fairly straightforward and is an adaptation of existing art.

**Review Confidence:**

4: The reviewer is confident but not absolutely certain that the evaluation is correct

---

### Decision · Program_Chairs · 2022-10-23

**Decision:**

Accept

**Comment:**

This paper gives a comprehensive study for the social bias problem in BERT in multiple languages. Although it is technically straightforward, its evaluation is sufficient and the results are interesting. I hope the authors can revise the paper based on reviewer comments.